# Electric Field-Driven Liquid Metal Droplet Generation and Direction Manipulation

**DOI:** 10.3390/mi12091131

**Published:** 2021-09-20

**Authors:** Jinwon Jeong, Sangkug Chung, Jeong-Bong Lee, Daeyoung Kim

**Affiliations:** 1Department of Mechanical Engineering, Myongji University, Yongin 449-728, Korea; jsungrobert@mju.ac.kr; 2Department of Electrical and Computer Engineering, The University of Texas at Dallas, Richardson, TX 75080, USA; jblee@utdallas.edu; 3Department of Information and Communication Engineering, Korea Army Academy at Yeong-cheon, Yeong-cheon 770-849, Korea

**Keywords:** gallium-based liquid metal, electro-hydrodynamic, electric fields, on-demand size controllable

## Abstract

A gallium-based liquid metal got high attention recently, due to the excellent material properties that are useful in various research areas. We report here on electric field-induced liquid metal droplet generation and falling direction manipulation. The well-analyzed electro-hydrodynamic method is a selectable way to control the liquid metal, as the liquid metal is conductive. The electric field-induced liquid metal manipulation can be affected by the flow rate (0.05~0.2 mL/min), voltage (0~7 kV), and distance (15 and 30 mm) between electrodes, which changes the volume of the electric field-induced generated liquid metal droplet and the number of the generated droplets. When the electric field intensity increases or the flow rate increases, the generated droplet volume decreases, and the number of droplets increases. With the highest voltage of 7 kV with 15 mm between electrodes at the 0.2 mL/min flow rate, the lowest volume and the largest number of the generated droplets for 10 s were ~10 nL and 541, respectively. Additionally, we controlled the direction of the generated droplet by changing the electric field. The direction of the liquid metal droplet was controlled with the maximum angle of ~12°. Moreover, we exhibited a short circuit demonstration by controlling the volume or falling direction of the generated liquid metal droplet with an applied electric field.

## 1. Introduction

An electro-hydrodynamic method for controlling the behavior of the liquid has been widely investigated [1,2]. Since this method can generate micro- and nano-sized droplets on-demand, it can be applied to various applications such as micro-scale structures [3,4,5,6], spray coating [7,8,9,10,11], heat transfer enhancement [12,13,14], fluid pumping [15,16,17,18], drying [19,20,21,22], and thrust [23,24,25]. Most of the electro-hydrodynamic methods were applied to printing applications [26,27,28,29,30]. An electric field-induced liquid ejection is mainly influenced by electric field intensity, gravity, capillary force, viscous force, and surface tension [31,32,33]. The dimensions of the liquid ejection system such as an orifice or a nozzle, the flow rate or solution properties also affect the dynamic behavior [32,33,34,35]. Especially, based on the electro-hydrodynamic (EHD) behavior of the liquid, a new printing system can overcome the limitations on the material type, geometry, and thickness of the receiving substrate [36,37]. Fundamental studies using pure diverse liquids to understand the electro-hydrodynamic behavior are well investigated [33]. There have been diverse electro-hydrodynamic studies for melting metals [37,38,39], but electro-hydrodynamic characteristics using liquid metal, which is the metal with a liquid phase at room temperature, has not been well investigated.

Among a variety of liquid metals, a gallium-based liquid metal has been utilized in various applications such as flexible [40,41] or wearable electronics [42,43], cooling material [44,45], tunable radio frequency (RF) antenna or metamaterial [46,47,48,49,50], bio-application [51,52] using the liquid metal’s non-toxicity, biocompatibility, relatively high electrical and thermal conductivities, infinite deformability, and large surface tension [53]. The locomotion and morphology manipulation of the liquid metal have been a challenging issue due to its oxidized surface [54]. We have reported a magnetic field-based manipulation using a permanent magnet or an electromagnet in an open space and microfluidic channel [55,56,57]. The electro-hydrodynamic manipulation of the liquid metal in aqueous solutions was investigated to understand the moving mechanism in the solution [58,59]. However, there is little study to understand the electro-hydrodynamic behavior itself, of the gallium-based liquid metal.

In this paper, we report on the liquid metal droplet generation and falling direction manipulation with an applied high electric field. We could control the volume of the generated liquid metal droplet by changing the electric field or liquid metal injection rate. Based on these studies, connecting a disconnected circuit using the volume control or falling direction manipulation of the generated liquid metal droplet was demonstrated.

## 2. Electric Field-Driven Liquid Metal Droplet Generation

A conceptual schematic of the electric field-based gallium-based liquid metal droplet generation is shown in Figure 1. A 250 μm thick insulator (PTFE, Alphaflon, Inc., Seoul, Korea) was coated on top of the circular-shaped copper (Cu) electrode, in order to prevent a corona discharge between the needle and the electrode [33,60]. We supplied the gallium-based liquid metal (Galinstan) with the flow rate ranging from 0.05 to 0.2 mL/min, using a syringe pump (LEGATO^®^ 380 EMULSIFIER, KD scientific, Inc.) from 15 mm above the Cu electrode. In order to generate gallium-based liquid metal droplets, a high DC voltage (3.5~7 kV) was applied using a voltage supply (DR450, Sciencestar, Inc., Seoul, Korea) between a circular-shaped Cu electrode (4 inch diameter, 1 mm thickness) and a needle (34 G plastic needle, DNV, Inc., Shenzen, China) with an inner diameter of 60 μm, which was electrically grounded. We have utilized an induction setup in which the charges are induced by an electric field under the configuration that the needle and the copper electrode are cathode and anode, respectively, since this setup has fewer safety concerns [61]. When the electric field is applied, the dynamic behavior of the liquid metal is determined by the competition between the electrostatic force due to the electric field and capillary force from the liquid’s viscosity and surface tension [33]. Therefore, the electrical charges on the surface of the induced liquid metal are pinched off the droplet. By applying a higher voltage, the smaller volume of liquid metal droplet was observed. In addition, the number of generated droplets was increased with the applied voltage. Two insets in Figure 1 show optical images when the liquid metal droplet was on the mode of ‘pinch off’ with and without the applied voltage, respectively. Moreover, the secondary satellite droplet was observed without the applied voltage, while the long filament shape was observed with the applied voltage. The electro-hydrodynamic behavior is influenced by several parameters such as the properties of the ejected liquid metal, specifications for the nozzle, applied voltages between the nozzle and electrode, and variation of the hydrostatic pressure by the syringe pump, which is related to the Poiseuille-type flow rate relation [61,62] as follows:(1)$$Q\;\sim \;\frac{\pi d_{n}^{4}}{128\mu L}\left(\frac{\varepsilon_{0}E^{2}}{2}-\frac{2\gamma }{d_{n}}+\Delta P\right)$$

From this equation, *Q* is the flow rate from the nozzle related to the formation of the micro-sized droplet, ∆*P* is the pressure drop between the piston and exit of the nozzle by the syringe pump, $d_{n}$ and *L* are the inner diameter and length of the nozzle, respectively, *μ* is the viscosity of the liquid metal in the syringe, $\varepsilon_{0}$ is the permittivity of the free space, *E* is the electric field, and *γ* is the surface tension regarding the interface between the air and liquid metal. The flow rate (*Q*) of the ejected liquid metal droplet from the nozzle is affected by the hydrostatic pressure (∆*P*), which means that the pressure of the liquid metal in the reservoir results from the syringe pump, the electric pressure ($\frac{\varepsilon_{0}E^{2}}{2}$), capillary pressure ($\frac{2\gamma }{d_{n}}$), and characteristics of the matter in the syringe.

In our experiment, as we controlled the injecting flow rate using a syringe pump, the force which presses the liquid metal vertically could be changed, resulting in the difference of the hydrostatic pressure. The height (*h*) between the needle and the electrode affected the magnitude of the electric field, which changed the ejecting flow rate [62,63]. Therefore, the injecting flow rate and height between the electrodes are important parameters to control the dynamic behavior of the liquid metal.

Figure 2 shows the balance of various forces which affect electrohydrodynamic jetting and Figure 3a illustrates consecutive time lapse images of the gallium-based liquid metal droplet generated by various applied voltages (0, 3.5, 5, and 7 kV) for ~22 ms taken at a shutter speed of 1/500 s (refer to Appendix A). In order to trace the falling track of the generated liquid metal droplet and compare the size of the droplets according to the applied voltages accurately, we recorded the overall sequences utilizing a high-speed camera (V 7.3, Phantom, Inc., New Jersey, USA). The gallium-based liquid metal droplet was supplied from a syringe at a constant speed of 0.2 mL/min. When a higher voltage was applied, an electric field was generated between the needle and electrode, resulting in the generation of a charged gallium-based liquid metal droplet whose electric charge was conserved temporarily at the interface of liquid/gas. Under the competition of the electric field force, surface tension, viscous force, and gas pressure, the meniscus was elongated gradually to form a Taylor-cone [64]. When the tangential force due to the electric field exceeded the surface tension, a droplet was generated (Figure 2). Therefore, the applied higher voltage can induce the larger charge accumulation on the surface, resulting in the smaller droplet and the larger number of generated droplets.

Different colors are used to clearly distinguish the generated droplets with different voltages (0 kV-black, 3.5 kV-blue, 5 kV-green, and 7 kV-red), in order to compare the falling path of each droplet. Figure 3b shows top-view images of the generated gallium-based liquid metal droplets. The measured falling distances of generated liquid metal droplets from the needle corresponding to the applied voltages of 0, 3.5, 5, and 7 kV were ~4.2, ~7.4, ~11.4, and ~13.7 mm, respectively during 22 ms. As shown in Figure 3, with the higher voltages, we obtained the longer falling distance and the smaller size of the liquid metal droplet (multimedia view).

Utilizing optical images of the falling gallium-based liquid metal droplet, the volume (nL) and number of the generated droplets for 10 s with different applied voltages were measured, as shown in Figure 4a,b, respectively. A diameter of the generated liquid metal droplet was measured through a pixel analysis using the image measurement software (ImageJ, Java). As the applied voltages were increased, the volume of the droplets was decreased, as shown in Figure 4a. In addition, with the increasing flow rate from 0.05 to 0.2 mL/min, the volume of the generated liquid metal droplet became smaller. However, the effect of the flow rate on the volume of the droplet was much smaller than that of the electric field. In contrast to the volume, the number of the generated droplets was increased with the applied voltages and the flow rate, as shown in Figure 4b. The volume and number of the generated liquid metal droplet(s) for 10 s at 7 kV with the flow rate of 0.2 mL/min were 10.0± 0.1 nL and 541, respectively.

Moreover, we investigated an effect of a height (*h*) on the volume and number of the generated liquid metal droplet(s), which was defined as a vertical distance from the needle and circular-shaped Cu electrode. In order to verify that the height between the needle and electrode impacts on the electro-hydrodynamic behavior of the generated liquid metal droplet, the equation regarding the electric field distribution corresponding to this system is introduced [65,66], as follows:(2)$$E=\frac{4V}{d_{n}\ln\left(\frac{8h}{d_{n}}\right)}$$
where *E* is the applied electric field at the tip of the nozzle, *V* is the applied voltage between the nozzle and electrode, $d_{n}$ is the inner diameter of the nozzle, and *h* is the distance between the nozzle and electrode. From this equation, we can verify that *h* is one of the key parameters which has a potential to make a difference in *E,* resulting in electro-hydrodynamic behavior of the liquid metal droplet, such as the volume and velocity of the droplet generation.

As shown in Figure 5, the square dot (dashed line) indicates the volume (black-colored) and number (red-colored) of the generated droplets for 10 s with a 15 mm distance, while the circle dot (solid line) indicates them with a 30 mm distance at a flow rate of 0.2 mL/min. As the height was decreased, the stronger electric field was induced. Consequently, with the lower height, the smaller droplet and additional droplets were generated. Compared to the volume of the droplet with *h* = 30 mm (47.0 ± 0.3 nL) under the applied voltage of 7 kV and flow rate of 0.2 mL/min, the volume is decreased to 10.0 ± 0.1 nL according to the change of *h* from 30 to 15 mm.

## 3. Electric Field-Driven Direction Manipulation

Based on studies of electric field-based liquid metal droplet generation, we investigated a way to control the direction of the generated liquid metal droplet by changing the electric field. Figure 6 depicts an on-demand falling direction control of the gallium-based liquid metal droplet using a ring-shaped Cu electrode [67]. The ring-shaped electrode was made of an acrylic mold (10 mm thickness, 20 mm inner diameter, 25 mm outer diameter) covered with a Cu tape whose depth is 0.1 mm. Various voltages (0, 3.5, 5, and 7 kV) were applied between the needle which was electrically grounded and the ring-shaped Cu electrode with 10 or 15 mm in height (*h*). Moreover, we defined the distance (*d*) as a horizontal displacement of the ring-shaped Cu electrode from the central axis of the needle. When *d* = 0 mm (Figure 6(b1)), the needle and the center of the Cu electrode is collinear. If the Cu electrode was moved horizontally +3.3 mm (right side) or −3.3 mm (left side) against the vertical central axis of the needle, the falling traces of the liquid metal droplet were changed (Figure 6(b2,b3)). In this system, the ring-shaped Cu electrode is utilized as not only the electrode which allows the ejected droplets to be charged, but also the deflection plate to manipulate the falling direction of the droplets. The deflection for the charged droplets is affected by various factors such as the electrical property of the charged droplets, resulting from the external electric field and geometric specifications related to the deflection plate [68]. If the center of the electrode which acts as the charge electrode and deflection plate is concentric with the needle, a symmetrical electric field is formed, which means that the left and right electrostatic force affecting the dynamics of the liquid metal droplet are in balance. However, as the electrode was moved in the left or right direction, which caused the asymmetrical electric field to be formed, the falling direction of the ejected liquid metal droplet was influenced by an unbalanced electrostatic force. The inset showed the angle (*θ*, *°*) between the center line of the needle and a line tangent to the trace of the generated liquid metal droplet.

Figure 7 shows consecutive time lapse images of the falling direction of the controlled-liquid metal droplet generated by different voltages (0, 3.5, 5, and 7 kV) (refer to Appendix A). Various colors (black, blue, green, and red) were added to optical images of each droplet in order to clearly distinguish the traces of the generated liquid metal droplets with different voltages (0, 3.5, 5, and 7 kV). We investigated the effect on height (*h*) between the needle and the ring-shaped Cu electrode, and distance (*d*), a horizontal displacement of the electrode against the vertical central axis of the needle. A flow rate was fixed at 0.2 mL/min. The position of the ring-shaped Cu electrode was manipulated using a translation stage (Edmund optics, Inc., Seoul, Korea). Under the initial condition (*h* = 10 mm, *d* = 0 mm), the gallium-based liquid metal droplet is generated vertically without any impact on the applied electric field. When the ring-shaped electrode was shifted to the left by 3.3 mm from the center, the trace of the generated liquid metal droplet was bent to the right relative to the central axis of the needle (Figure 7b). Similarly, the trace of the droplet was changed to the left as the electrode was moved to the right (Figure 7c). In addition, with the increasing voltages from 0 to 7 kV, the angle of the trace of the generated liquid metal droplet was increased, as shown in Table 1. When the height (*h*) was decreased, the stronger electric field was induced resulting in the larger angle of the trace (Figure 7c,d). We confirmed that the slight differences of the angle of the trace between the left and right directions for 3.5, 5, and 7 kV at *h* of 10 mm have occurred. This would be based on the magnitude difference of the electric field as we finely control (tenth of millimeter scale) the position of the electrode by hand.

In addition, we changed the electric field by covering the electrode with a rubber plate to further investigate the electro-hydrodynamic behavior of the liquid metal droplet generation. Figure 8 illustrates the conceptual schematic and optical images of the direction controlled-liquid metal droplet by covering the electrode with variously applied voltages. In order to control the area of the electrode, the rubber plate which served as an insulator was utilized (multimedia view). In Figure 1, the PTFE layer was utilized in order to prevent the electrical arc (so called ‘corona discharge’) [12]. Unlike the objective of using the insulating layer in Figure 1, we utilized the rubber pad to make the electric field weaken by covering a certain part of the Cu electrode. A 4 inch circular-shaped Cu electrode with a hole (diameter of 5 mm) was used. An electrode holder (12.5 (*W*) × 12.5 (*D*) × 0.5 (*H*) cm^3^) was fabricated by a 3D printer (DP 200, Sindoricoh, Inc., Seoul, Korea) using poly lactic acid (PLA) to stably support the electrode and facilitate the movement of the rubber plate. Optical images of the experiment apparatus are shown in Figure 8(b1,b2). The flow rate was set at 0.2 mL/min and the height from the needle to the circular-shaped Cu electrode was 15 mm. Without the rubber plate, the gallium-based liquid metal droplet that ejected from the needle fell vertically (Figure 8(a1,c4)). However, once the rubber plate covered the right side of the electrode by ~43% of the electrode area, the generated gallium-based liquid metal droplet was bent in the left direction (Figure 8(a2,c1–c3)) (refer to Appendix A). Moreover, when the rubber plate covered the left side of the electrode, the trace of the liquid metal droplet generation was bent to the right (Figure 8(c5–c7)). As the voltages applied between the needle and electrode were increased, the angle of the trace of the generated liquid metal droplet was gradually increased. Table 2 shows the angles of the trace of the generated liquid metal droplet against the covered area and various voltages.

## 4. Controlled Short Circuit Demonstration

Based on fundamental electro-hydrodynamic studies of the gallium-based liquid metal droplet generation, we demonstrated the shorting of an open circuit utilizing the on-demand volume and falling direction control of the liquid metal, as shown in Figure 9 and Figure 10.

An experimental setup is shown in Figure 9a. Here, 7 kV was applied between a needle and 4 inch circular-shaped Cu electrode (5 mm diameter hole). The height was 15 mm, and the flow rate was 0.2 mL/min. We prepared a printed circuit board (PCB) with two light emitting diodes (LEDs) in red and green color. The optical image and circuit diagram were illustrated in Figure 9b,c, respectively. An electrical circuit was composed of two LEDs (red and green) and an electrode made of conductive ink. The red and green LEDs were linked to both switch 1 and 2 whose gaps were 1 and 2 mm, respectively. Once the droplet was generated, we blocked the other generated droplets to remove any damage of the circuit. Without applying the higher voltage, the larger volume of the liquid metal droplet was generated and fell straight down to the electrical circuit, which was big enough to make contact with both switch 1 and 2, resulting in turning on both the green and red LEDs (Figure 9(d1,e1)). However, with an applied voltage of 7 kV, a smaller liquid metal droplet was dropped, which could close switch 1 and selectively result in turning on the red LED only (Figure 9(d2,e2)) (refer to Appendix A).

Moreover, we applied the concept of controlling the angle of the falling liquid metal droplet by switching, as shown in Figure 10a (refer to Video Appendix A). The ring-shaped Cu electrode with the same specification (Figure 6) was utilized. We fabricated two electrical circuits with two different LED colors (yellow and green). An optical image of the fabricated electrical circuit is shown in Figure 10b. A switch incorporated with the circuit, which consisted of multiple electrodes 0.6 mm wide and 0.4 mm gap, was shown in an inset. If 7 kV was applied between the ring-shaped electrode and the needle and the electrode was moved 3.3 mm to the right from the central axis of the needle, the gallium-based liquid metal droplet fell to the left circuit resulting in turning on the yellow LED (Figure 10c,e). Similarly, the generated liquid metal droplet fell to the right circuit and the green LED was turned on, resulting from the connection of the switch when the ring-shaped electrode was moved 3.3 mm to the left direction under 7 kV (Figure 10d,f).

## 5. Conclusions

In this paper, we investigated the electro-hydrodynamic behavior of the gallium-based liquid metal. The liquid metal droplet volume decreased, and the generated droplet number increased with the higher electric field and flow rate. The falling direction of the generated liquid metal droplet was controlled by changing the electric field using a ring-shaped electrode. When the ring-shaped electrode was moved from the central axis, the trace of the generated liquid metal droplet was also bent accordingly. Additionally, the trace of the generated liquid metal droplet was also controlled by covering the electrode with a rubber plate, which changed the electric field. Finally, we turned on the LED by controlling the volume and falling direction of the generated liquid metal droplet. The generation and direction control of the liquid metal droplet by the electro-hydrodynamic method can be applied to the liquid metal inkjet and 3D shape form of the micro-sized liquid metal.

## Figures and Tables

**Figure 1 micromachines-12-01131-f001:**
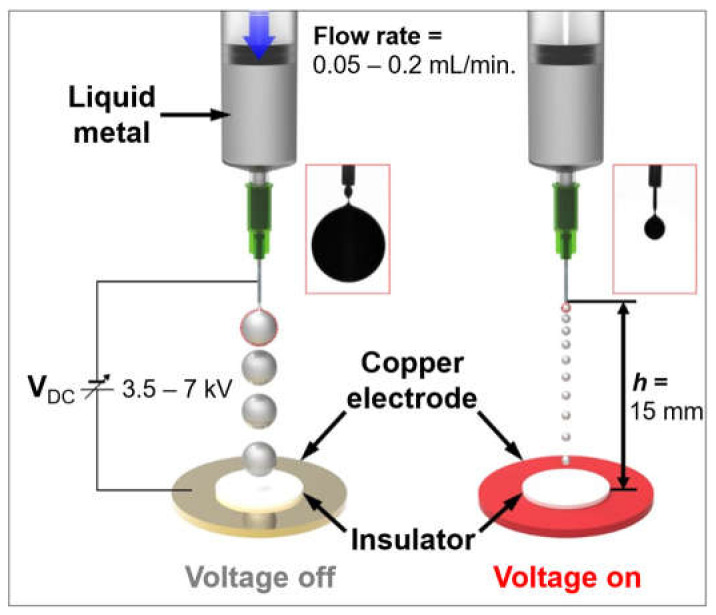
Schematic of electro-hydrodynamic gallium-based liquid metal droplet generation according to the applied high voltage between a needle and circular-shaped Cu electrode.

**Figure 2 micromachines-12-01131-f002:**
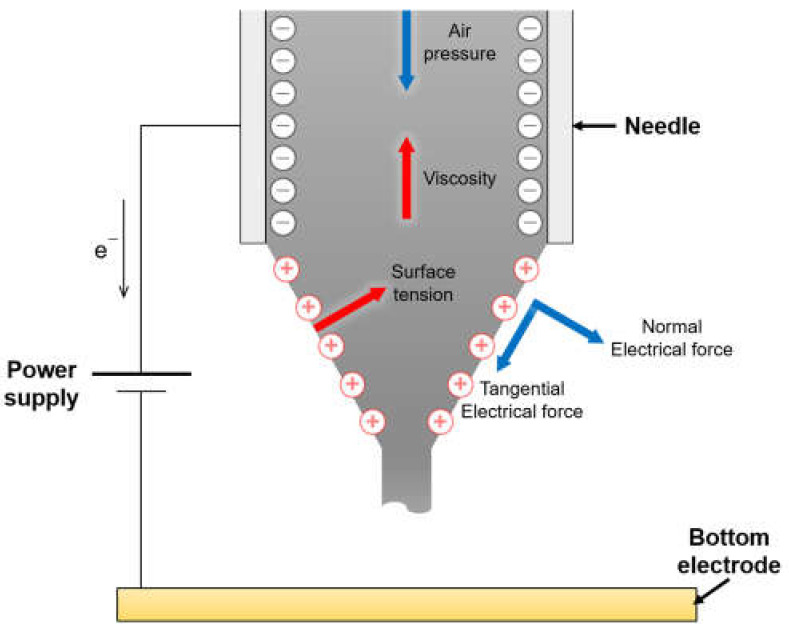
Schematic diagram of the force balance between pneumatic force, viscosity, surface tension, and electrical force during electro-hydrodynamic jetting.

**Figure 3 micromachines-12-01131-f003:**
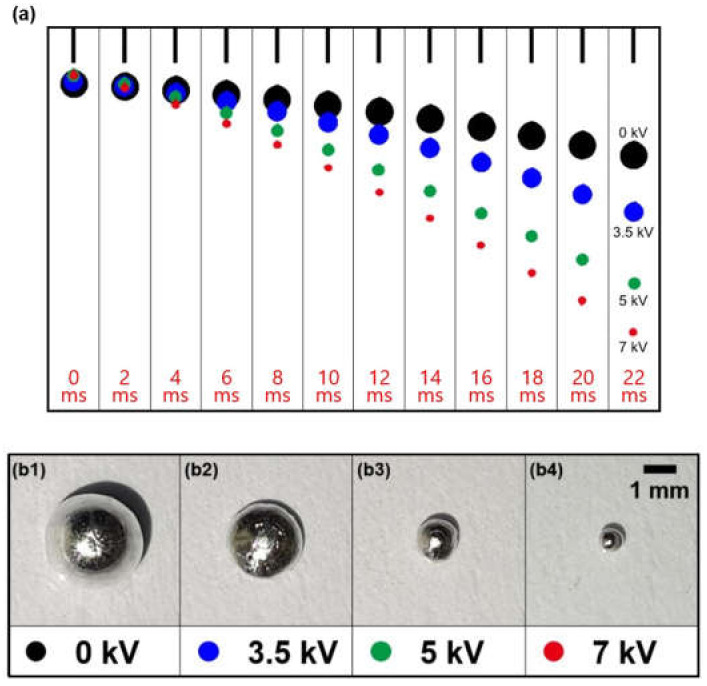
(**a**) Superimposed sequences of the falling gallium-based liquid metal droplet and (**b**) optical images of dropped liquid metal droplet, generated by the electro-hydrodynamic method with various applied voltages of (**b1**) 0, (**b2**) 3.5, (**b3**) 5, and (**b4**) 7 kV between a needle and circular-shaped Cu electrode (multimedia view).

**Figure 4 micromachines-12-01131-f004:**
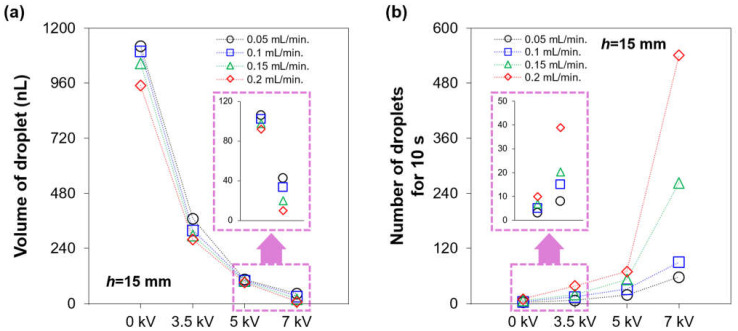
Impacts of the applied voltages and flow rate on the (**a**) volume and (**b**) number of the generated gallium-based liquid metal droplets for 10 s.

**Figure 5 micromachines-12-01131-f005:**
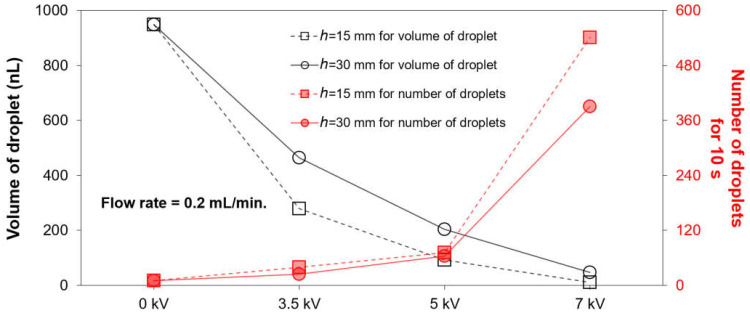
Impacts of height between the needle and circular-shaped Cu electrode on the volume and number of the generated gallium-based liquid metal droplet(s) ejected from the needle at the flow rate of 0.2 mL/min, by applying various voltages (0, 3.5, 5, and 7 kV).

**Figure 6 micromachines-12-01131-f006:**
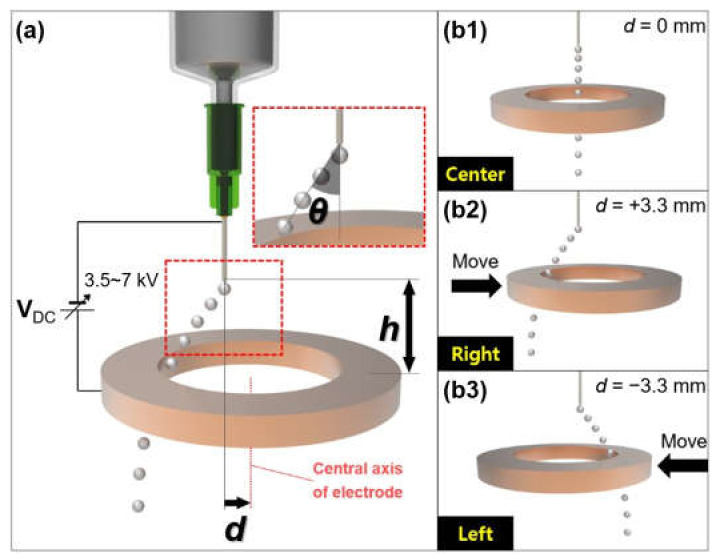
On-demand falling direction manipulation of the gallium-based liquid metal droplet generated by the electro-hydrodynamic method using a ring-shaped Cu electrode: (**a**) Schematic of overall system regarding direction control of liquid metal droplet; (**b1**) falling of droplet in the center direction under *d* = 0 mm; (**b2**) falling of droplet in the left direction under *d* = +3.3 mm; (**b3**) falling of droplet in the right direction under *d* = −3.3 mm.

**Figure 7 micromachines-12-01131-f007:**
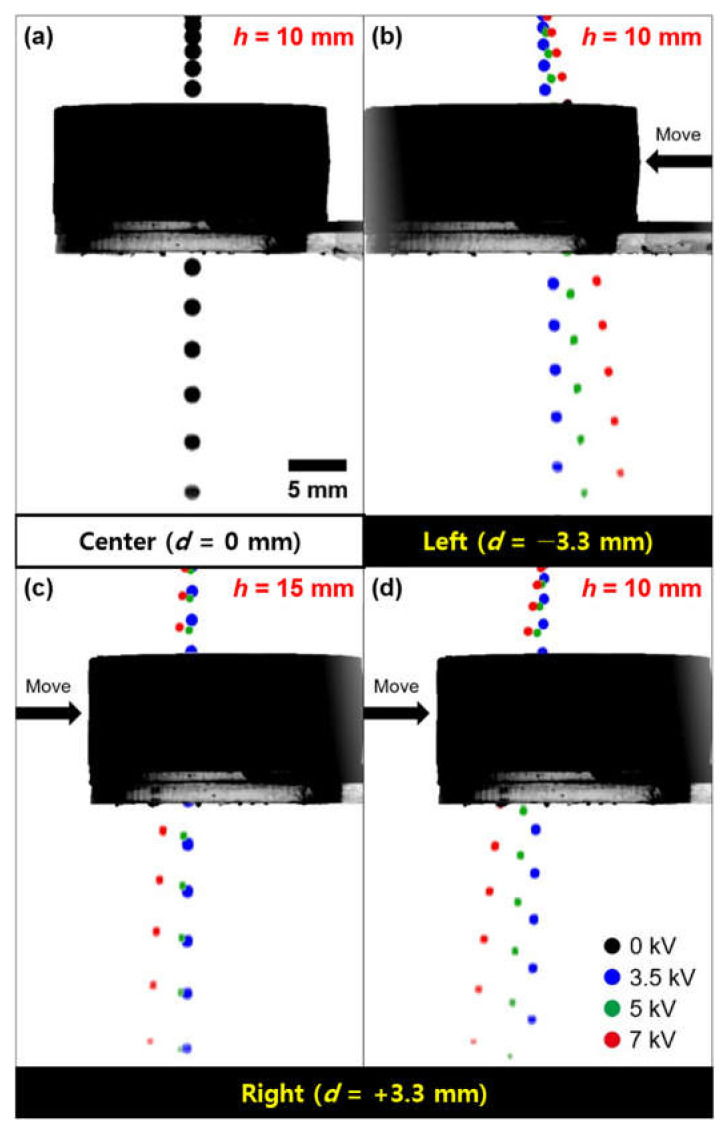
Superimposed sequences for the on-demand direction manipulation of the falling gallium-based liquid metal droplet by applying various voltages (0, 3.5, 5, and 7 kV) according to a position of the ring-shaped Cu electrode (multimedia view): (**a**) falling of droplet in the center direction at *h* = 10 mm; (**b**) falling of droplet in the left direction at *h* = 10 mm; falling of droplet in the right direction at (**c**) *h* = 15 mm and (**d**) *h* = 10 mm.

**Figure 8 micromachines-12-01131-f008:**
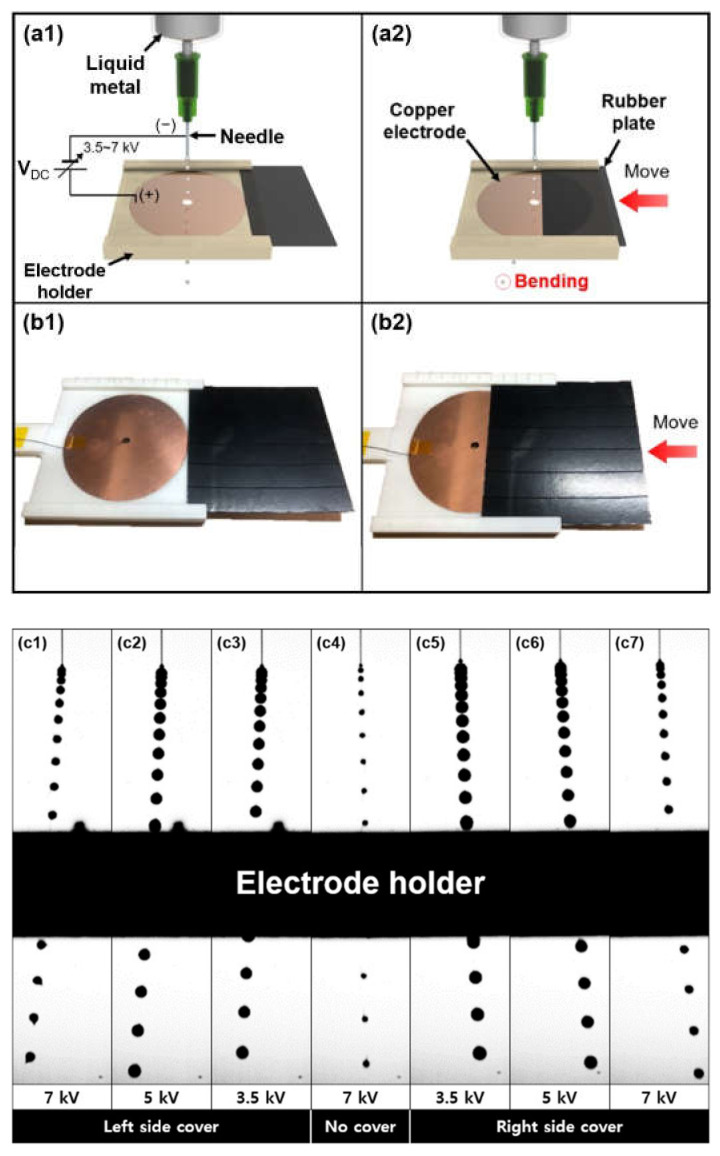
(**a**) Conceptual schematics and (**b**) experiment apparatus on the falling direction control of the gallium-based generated liquid metal droplet by regulating an area of the circular-shaped electrode affected by an electric field: (**a1**, **b1**) uncovered electrode and (**a2**, **b2**) covered electrode with rubber plate; (**c**) superimposed sequences for the area of the electrode-dependent falling liquid metal droplet according to various high voltages (3.5, 5, and 7 kV) (multimedia view): (**c1**–**c3**) falling of droplet in the left direction with left side cover; (**c4**) falling of droplet in the center direction with no cover; (**c5**–**c7**) falling of droplet in the right direction with right side cover.

**Figure 9 micromachines-12-01131-f009:**
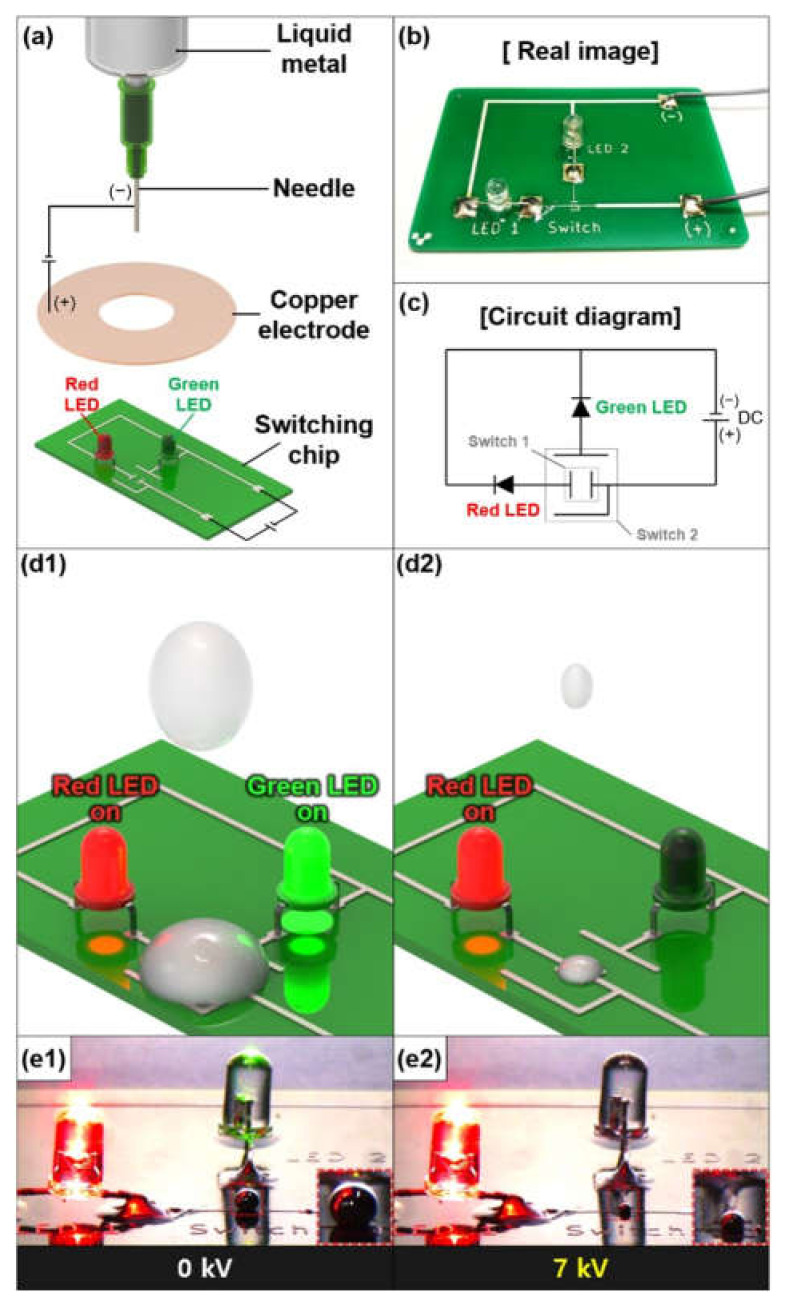
(**a**) Conceptual schematic of short circuit demonstration based on a volume control of the gallium-based liquid metal droplet using a circular-shaped Cu electrode with a hole and electrical circuit; (**b**) optical image; (**c**) diagram of fabricated electric circuit; (**d1,d2**) conceptual schematic; and (**e1,e2**) optical images of the switching application with and without 7 kV (multimedia view).

**Figure 10 micromachines-12-01131-f010:**
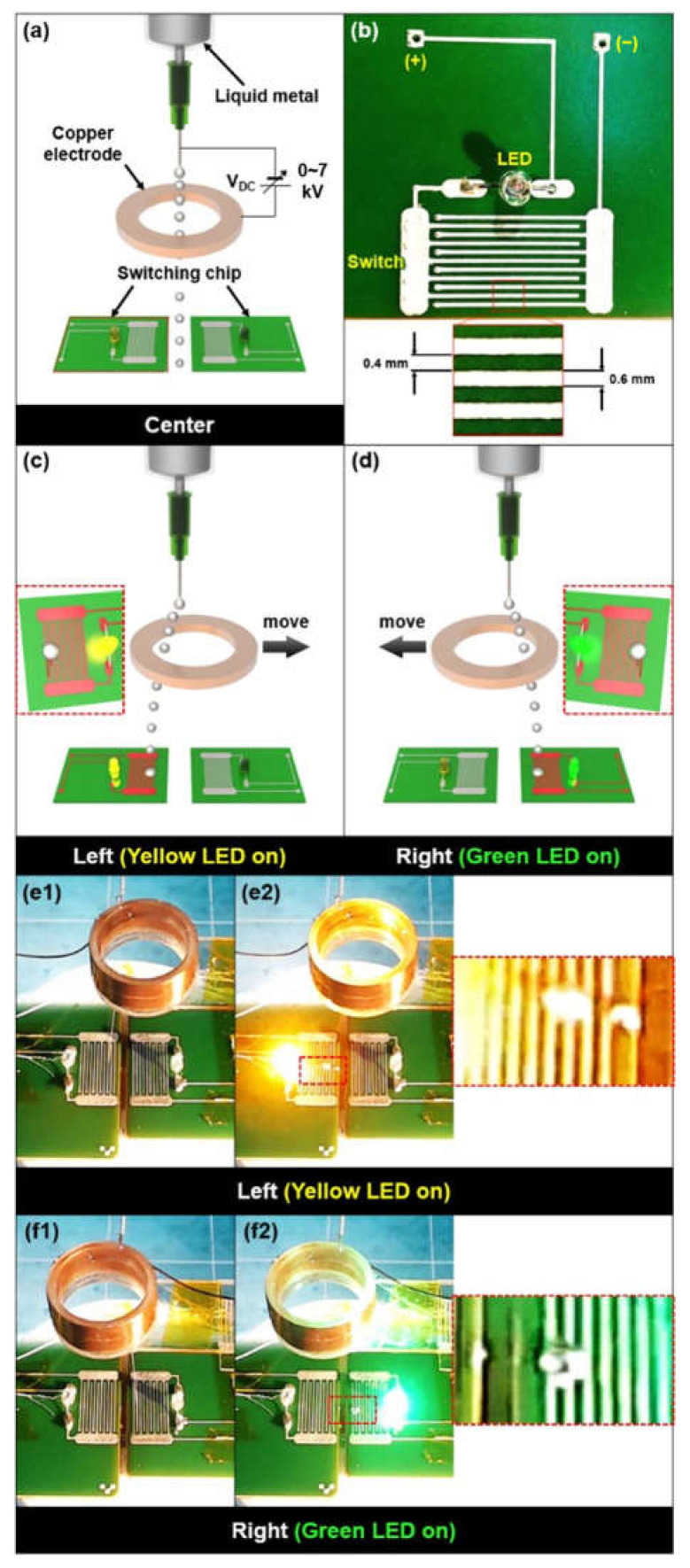
(**a**,**c**,**d**) Conceptual schematics and (**e**,**f**) optical images of short circuit demonstration using (**b**) an electrical circuit based on the falling direction control of a gallium-based liquid metal droplet depending on a position of a ring-shaped electrode (multimedia view): Yellow LED (**e1**) off and (**e2**) on by falling of liquid metal droplet on the electrical circuit; Green LED (**f1**) off and (**f2**) on by falling of liquid metal droplet on the electrical circuit.

**Table 1 micromachines-12-01131-t001:** Angle of the trace on different voltages and positions.

Position	Angle of the Trace (°)		
	3.5 kV	5 kV	7 kV
Left (*h* = 10 mm)	1.6	6.5	12.8
Right (*h* = 10 mm)	1.1	4.6	11.4
Right (*h* = 15 mm)	0.6	1.7	5.6

**Table 2 micromachines-12-01131-t002:** Angle of the trace on different voltages and covered areas.

Covered areas of Electrode	Angle of the Trace (°)		
	3.5 kV	5 kV	7 kV
Left	2.7	3.9	5.1
Right	2.9	4	5.2

## Data Availability

Not applicable.

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
