# Peer review of "Electric Field-Driven Liquid Metal Droplet Generation and Direction Manipulation"

_micromachines, 2021, doi:10.3390/mi12091131_

Round 1
Reviewer 1 Report
Page 3. Line 115 “When the tangential force due to the electric field exceeds the surface tension, a droplet was generated”, this sentence is not so clear to me in terms of the force balance and the droplet formation. If standard in the field please make a reference else a figure would be helpful.
Page 4. Line 135 “As applied voltages were increased, volume of the droplets was exponentially decreased, as shown in Fig. 3(a).” I do see from the data presented in the figure that the dependence is very strong. If exponential as claimed by the authors then a plot where y axis is logarithmic (and x-axis unchanged) should show a straight line. If this is done and the straight line is borne out it would support the claim of an exponential result. However should not be surprising if the exponential behavior is not there, but instead a power law underlies the data (to be obtained by plotting on double log and finding a straight line) Analogous systems where instead of voltage, external flow rates are used, power law dependence have been reported which is easier to explain theoretically. For instance in: “Examining the Effect of Flow Rate Ratio on Droplet Generation and Regime Transition in a Microfluidic T-Junction at Constant Capillary Numbers” (Inventions 2018, 3, 54) and in “Dripping to Jetting Transitions in Coflowing Liquid Streams” (PRL 99, 094502 (2007)). The paper would benefit from some discussion on the precise functional form if the data permit this. If a power law gives a better fit I suggest to rewrite the conclusions and discussion accordingly.
Reviewer 2 Report
The paper is a very important application of droplet atomization and the effect of various factors into droplet size, spacing , speed, and impingement.
- The structure of the paper makes it very hard to follow, the structure should be introduction, methods and data analysis, results, discussion, and conclusion. The authors studied various conditions and had the methods, results, discussion for all per condition separation is needed to recreate the study.
- The study lacks statistical treatment was the experiment repeated or the results are consistent within the experiment? All the figures lack the standard deviation, does the droplet size change with location within the experiment?
- The paper relies on graphical representation using image processing and super imposing the results this is very good though it must be summarized in tables, plots, …etc.
- The discussion section of the paper needs to be improved the results are they in agreement with theory or expectation why one side is different that the other do the results follow a trend and have anyone reported those trends before, ...etc the discussion is a very important part of archival papers otherwise the paper is an experiment report
Comments specific for the paper:
- In the paragraph from lines 85-93 the term ε_0 appears is it ε0?
- In the paragraph from lines 153-162 the term d_n appears is it dn?
- Table 1 the angles between the left and right direction varies for the 10 mm condition:
- Is that variation significant or not what type of statistical treatment was made to come up with those conclusions?
- If the variation id significant why is there a change in the results?
- There are various correlations presented in this study did the authors correlate the findings to theoretical expectations? If so what is the deviation or bias between the study results and the theoretical outcome
- Did the droplet diameter vary from one experiment to another i.e. when the electrode location was changed did the diameter stayed the same especially close to the electrode?
- The use of the multimedia is a good representation of the accuracy though the dimensions of said systems need to be on the schematics for repeatability
Reviewer 3 Report
The article “Electric Field-Driven Liquid Metal Droplet Generation and Direction Manipulation” is focused on the production of polymeric fiber. In principle I find interesting the object of the research, but I think that the article could benefit of a process of major revision, reducing the emphasis on the novelty of the system proposed, as described point by point in the following:
- Probably the authors could improve the first part of the introduction, related to the EHD printing properties, introducing some references about the dimension of the dots or fibers that could be achieved using the EHD printing. The period form line 37 to line 39 is not clear, it seems that authors are claiming that the dimension of the nozzle are related to the printing properties and, because this is a very important point in all the EHD systems it will be better to explain the advantages and/or disadvantages. In fact, authors don’t introduce a reference to alternative EHD printing system that, avoiding the nozzle, could allow the printing of high viscous materials (see Lab on a Chip, 2016, 16(2), 326–333, advances in polymer technology Volume2020 Article Number1252960 DOI10.1155/2020/1252960 and references therein) still difficult to manipulate using nozzles or orifices. Moreover authors claim that “electro-hydrodynamic characteristics using liquid metal which is liquid phase at room temperature has not been well investigated”, but some experiments still exists (2016 Advanced Functional Materials 26(6), pp. 833-840 ).
- Paragraph 2: “By applying a high voltage, the smaller volume of liquid metal droplet was ” How was demonstrated this relation? please add the value you refer for high voltage and smaller volume. What do you mean for bias?
- Figure 2(a) illustrates consecutive time lapse images of gallium-based liquid metal droplet generated with variously applied voltages but it is not clear how was measured the distance occupied form the flying droplet. The nozzle and the bottom electrode could be moved? Moreover from the results, bigger volume droplets could travel for short distance when compared with small volume droplet even if the trajectory doesn’t change. How many time was repeated the image acquisition? Please add error bars.
- Figure 2a represent quite big droplets, I suppose that these droplets were generated in period longer than 20 ms, please clearify.
- Figure 3: at 0 kV a droplet of about 900-1200 nL is generated, but if the voltage is zero the jetting is not yet started. I observe the same problem in figure 4.
- In principle I find interesting the manipulation of the jetting using the electric field but for real case of use I find very cumbersome the set-up and the required movement of the ring shaped electrodes that should be moved for controlling the jetting. Probably this would add a limitation of the set-up and in the control of the distance between the nozzle and the target and finally the movement of the nozzle itself would be easier.
- I find really confusion the representation of the jetting a zero voltage (again in Figure 6a where the black colour stand for the 0 kV)
- English should be revised all over the text. Some recurring words are still present in the manuscript.
